# Simulation and Experimental Study of Dynamical Recrystallization Kinetics of TB8 Titanium Alloys

**DOI:** 10.3390/ma13194429

**Published:** 2020-10-05

**Authors:** Wenwei Zhang, Qiuyue Yang, Yuanbiao Tan, Min Ma, Song Xiang, Fei Zhao

**Affiliations:** 1Guizhou Key Laboratory of Materials Mechanical Behavior and Microstructure, College of Materials and Metallurgy, Guizhou University, Guiyang 550025, China; Zww19960916@163.com (W.Z.); yangqy979@126.com (Q.Y.); sxiang@gzu.edu.cn (S.X.); Fzhao@gzu.edu.cn (F.Z.); 2College of Materials Science and Engineering, Hunan University of Science and Technology, Xiangtan 411201, China

**Keywords:** TB8 titanium alloy, hot compression, DRX behavior, FEM

## Abstract

The dynamic recrystallization (DRX) behavior in the hot working of TB8 titanium alloy was studied by using the experiment and finite element simulation (FEM) method. The results showed that the DRX behavior of TB8 titanium alloys was drastically affected by the hot processing parameters. The rising deformation temperature and reducing strain rate led to an augmentation in the grain size ($d_{DRX}$) and volume fraction ($X_{DRX}$) of DRX grains. In view of the true stress–strain curves gained from the experiment, the $d_{DRX}$ and $X_{DRX}$ models of DRX grains were constructed. Based on the developed models for DRX of TB8 titanium alloy, the isothermal forging process of the cylindrical samples was simulated by the DEFORM-3D software. The distributions of the effective strain and $X_{DRX}$ for DRX were analyzed. A comparison of the $d_{DRX}$ and $X_{DRX}$ of DRX grains in the central regions of the samples between the experimental and FEM results was performed. A good correlation between the experimental and simulation results was obtained, indicating that the established FEM model presented good prediction capabilities.

## 1. Introduction

It is well known that the aim of metal forming is primarily to obtain components with excellent mechanical properties, in addition to the shape and dimension of the parts. The mechanical properties of the parts are remarkably associated with the microstructure formed during hot working, which is affected by hot processing parameters. The in-depth investigation of the microstructure evolution of metals during hot working is very helpful to obtain the optimal processing parameters of hot working. DRX is generally considered to be a dominant phenomenon during the microstructure evolution of metals, which can significantly refine the grain size of metals and enhance the mechanical property of metals. Thus, it is quite critical to reveal the DRX behavior of metals during hot working for the control of the final microstructures of the components. 

Recently, a great number of investigations into the kinetic behavior of the DRX of metals has been carried out [1,2,3]. Nevertheless, the process of DRX is a continuous dynamic course, which significantly increases the difficulty of dynamically analyzing the evolution of DRX during hot working. With the unprecedented advancement of computer technology, the software of FEM has been extensively applied to reveal the microstructural evolution of metals during hot working [4,5,6,7,8,9,10,11]. OuYang et al. [4] established the DRX kinetics model and revealed the nucleation mechanism of the DRX of Ti-10V-2Fe-3V in β processing. The bulging mechanism was considered to be a main mechanism of DRX for Ti-10V-2Fe-3V during hot compression, and the $X_{DRX}$ of DRX gradually rose with the increment of strain and temperature, as well as the decrease in strain rate. Li et al. [6] investigated the DRX behavior in AZ80 magnesium alloy via the coupling of DEFORM-3D, along with cellular automaton (CA) based on experimental data. The model of DRX in AZ80 alloy was constructed, and the simulation results of the DRX process are very consistent with the experimental results. Donati et al. [9] predicted the general trend of recrystallized grain size ($d_{DRX}$) evolution in a real industrial case with 3D FEM simulations, and the results illustrated that the constructed model can well predict the general trend of $d_{DRX}$ evolution in the regions characterized by low strain, but nevertheless the model diverges in the regions characterized by severe strain. Quan et al. [10] implanted the kinetics model of DRX into the DEFORM-3D software to analyze dynamically the evolution process of DRX in the AlCu4SiMg alloy during hot compression. It was found that the volume fraction of DRX rose with the reduction in the strain rate and augmentation of the deformation temperature, which was completely in line with the results obtained from the experiment observations. Wu et al. [11] established a physically-based constitutive model, considering the relationship between the stress and the dislocation, and incorporated this model into ABAQUS software to illustrate the microstructure evolution of AZ61 alloy during hot compression. The result shows that the microstructure evolution of AZ61 alloy, including DRX fraction, grain size and grain shape, can be reliably simulated by using the developed FEM model. To sum up, the FEM has been widely considered to be an effective way to predict the microstructural evolution of metals, and optimize the process parameters during hot working.

TB8 titanium alloy presents ultra-high strength, good ductility and exceptional corrosion resistance, which have been extensively employed to manufacture aircraft fasteners in the aerospace industry [12,13,14]. The processing window of TB8 titanium alloy is quite narrow during hot working, owing to the generation of a larger deformation resistance for TB8 titanium alloy with an ultra-high strength. So as to boost the hot-workability, as well as the mechanical properties of TB8 titanium alloys, it is quite meaningful to deeply understand the microstructural evolution of TB8 titanium alloys. However, few efforts have been made to investigate the microstructural evolution of TB8 titanium alloys by the FEM method, especially the evolution of DRX. Thus, it is pressing to study the coupling of the DRX kinetic model to the FE model for the hot working of TB8 titanium alloys, in order to achieve the control and optimization of the final microstructure of the components.

The objective of the present paper is to construct a kinetics model of the DRX of TB8 titanium alloys via the process of hot working based on the experiment data, and then implant the established kinetics model of DRX into the FE model to characterize dynamically the DRX behavior of the TB8 titanium alloys during hot compression tests. The effects of deformation parameters on the $X_{DRX}$ and $d_{DRX}$ were analyzed and discussed. A comparison between the experimental results and simulated results has been performed to verify the feasibility of the simulated results. 

## 2. Experiment

The material employed in this paper was a forged TB8 titanium alloy with a diameter of 60 mm, and the composition of TB8 was shown in Table 1. The temperature of β→α + β transformation was measured to be 815 °C [14]. The received alloy was solution-treated at 900 °C, soaked for 30 min in high vacuum and subsequently cooled to room temperature in water. Figure 1 depicts the original microstructure of the heat-treated samples, which was composed of equiaxed recrystallized grains of β phase with an average grain size of approximately 87 µm.

Cylindrical samples with ø8 × 12 mm were prepared from the heat-treated bars. To thoroughly understand the DRX evolution of TB8 titanium alloy, the hot compression test was operated on a Gleeble 3500 (Dynamic Systems Inc. (DSI), Poestenkill, NY, USA). The testing temperature was varied from 850 to 1000 °C and the strain rate was varied from 0.001 to 1 s^−1^. The samples were heated to a given temperature at a rate of 20 °C/s for 5 min so that the temperature in the center of the deformed samples was the same as that on the surface of the deformed samples prior to deformation, and then all samples were compressed to a true strain of 0.8. To keep the deformed microstructure at the elevated temperature, the deformed sample was immediately water-quenched to 25 °C. Moreover, some samples were selected to be deformed to 0.2, 0.4, 0.6 and 0.8 under the hot processing parameters of 900 °C/0.001 s^−1^, respectively, for analyzing the effect of strain on the DRX behavior of TB8 titanium alloy. In order to examine the microstructural evolution of TB8 titanium alloy during hot compression by using a Leica DMI5000M (Leica Instruments GmbH, Wetzlar, Germany), all deformed samples were cut along the radial direction of the deformed samples. For optical observation, the deformed samples were grinded, mechanically polished and chemically etched using a Kroll reagent of 10 mL HF, 30 mL HNO_3_ and 60 mL H_2_O. To analyze the characteristic of grain boundaries during compression deformation by using an Electron Backscatter Diffraction (EBSD, Oxford Instruments, Abingdon, UK) technique, EBSD samples were ground and electro-polished in a solution of 70 mL CH_3_OH, 10 mL HClO_4_ and 20 mL (CH_2_OH)_2_ at the voltage of 20 V for 10 s. EBSD detection was performed at the voltage of 20 keV. The step size for EBSD observation was set as 0.8 µm.

## 3. Results and Discussion

### 3.1. The Flow Behavior and Deformed Microstructure

Figure 2 illustrates the flow curve of deformed samples at different hot working parameters. It was found that the flow behavior was dramatically influenced by the deformation parameters. At the strain rate lower than 0.1 s^−1^, the flow stress rapidly rose to a peak stress with the increase of strain, then hardly changed, showing that flow curves present a characterization of dynamic recovery. This is attributed to the deformation time being too short to permit the dynamic recrystallization at a high strain rate. When the strain rate was less than 0.1 s^−1^, a typical characterization of DRX was visible. At a given strain rate, it was found that the flow stress was lowered with the augmentation of temperature. This is associated with the promotion of grain boundary mobility with the elevation of the temperature during hot working.

To expound the DRX behavior of TB8 titanium alloys, the typical deformed microstructures of TB8 titanium alloys after hot deformation were observed, as depicted in Figure 3. It was observed from Figure 3a that only a few fine recrystallized grains formed in the local region, indicating that the deformation mechanism is mainly dynamic recovery under this deformation condition. With rising temperatures and a reduced strain rate, the number and size of recrystallized grains gradually increased (Figure 3b–f). At deformation parameters of 900 °C/0.001 s^−1^, a full dynamic recrystallization was observed (Figure 3g). With further increases in the deformation temperature, the recrystallized grains were obviously becoming coarser (Figure 3h). The effects of deformation temperature and strain rate on the DRX behavior of TB8 titanium alloys were quantitatively characterized, as illustrated in Figure 4 and Figure 5. It was found from Figure 4 that the $X_{DRX}$ was less than 25% at the strain rate higher than 0.1 s^−1^ for all testing temperatures. When the strain rate was less than 0.1 s^−1^, the $X_{DRX}$ augmented rapidly to above 55%. This is due to the fact that the process of hot working is a heat activation process, and thus the dynamic recrystallization needs enough time to proceed during hot working. It was seen from Figure 5 that the $d_{DRX}$ slightly augmented with the rising deformation temperature at a strain rate higher than 0.1 s^−1^, while it dramatically increased at a strain rate less than 0.1 s^−1^. This is attributed to the fact that the grain boundary and subgrain boundary are easily migrated at higher deformation temperatures and lower strain rates, which promotes the coarsening of recrystallized grains.

So as to further reveal the effects of strain on the DRX behavior of TB8 titanium alloys, the EBSD grain maps of the TB8 titanium alloy deformed at different strains are depicted in Figure 6. It is seen from Figure 6a that only a small number of fine recrystallized grains were visible at a true strain of 0.2. When the strain rose from 0.2 to 0.8, the $X_{DRX}$ and $d_{DRX}$ augmented obviously (Figure 6b–d). This is owing to the fact that the increasing strain leads to the augmentation of the deformation storage energy, which promotes the occurrence of DRX during hot working. Figure 7 illustrates the EBSD IQ maps overlaid with the grain boundaries of the TB8 titanium alloy deformed at different strains. The blue lines are on behalf of low angle grain boundaries (2–15°) and the red lines are representative of high angle grain boundaries (>15°). In general, the formation of low angle grain boundaries (LAGBs) in the interior of grains resulted from plastic deformation during hot working. It can be observed from Figure 8 that the misorientation angles for the LAGBs were mainly distributed in the angle range of 2–10°. A quantitative analysis in the number fraction of LAGBs was performed, as depicted in Figure 9. It is seen that the number fraction of LAGBs was firstly increased with a rising strain from 0.2 to 0.4, and then gradually decreased with the augmentation of strain. Liu [15] and Sachtleber et al. [16] reported that the density of dislocation inside the grain was associated with the misorientation angle across dislocation boundaries and the area of dislocation boundary per unit volume. To more quantitatively analyze the change in the dislocation density with true strain, the value of $\frac{1}{S}\sum \theta \times lb$ was used to approximately characterize the dislocation density in the interior of the grains [17]. θ is the behavior of the misorientation angle across dislocation boundaries. $l_{b}$ and S are the measured step size and the measured area for EBSD observation, respectively. As illustrated in Figure 10, the density of dislocation in the grain’s interior significantly increased with the augmentation of strain from 0.2 to 0.4. This shows that the increasing strain results in an increase in the density of dislocation. It is worth noting that the density of dislocation is slowly decreased with the further increasing strain from 0.4 to 0.8. The reason for this fact is that the increase in deformation storage energy with the strain promotes the increase in the $X_{DRX}$ during hot working, which causes the consumption of a great deal of dislocation. 

### 3.2. Peak Strain and Critical Strain

During hot working, the flow stress quickly rose to a peak value with the rising true strain at the start stage of the deformation, and then slowly decreased to a steady flow stress. The peak strain ($\varepsilon_{p}$) corresponding to the peak stress ($\sigma_{p}$) can be obtained from Figure 2. In general, the dislocation density is quickly increased with the rising strain at the starting stage of deformation. The dynamic recrystallization can occur before the stress reaches the $\sigma_{p}$ due to the dislocation accumulation [18]. The strain corresponding to the starting point of DRX is the critical strain ($\varepsilon_{c}$) during hot deformation. For the onset point of DRX, an inflection point (namely ∂(∂θ/∂σ)/∂σ=0) can be attained in the curve of the strain hardening rate (θ) with true stress (σ), which can be used to measure the value of $\varepsilon_{c}$ corresponding to the critical stress ($\sigma_{c}$) at an inflection point [19,20]. In this paper, the values of $\varepsilon_{p}$ and $\varepsilon_{c}$ at different deformation parameters were measured, as listed in Table 2. 

It is seen that the values of $\varepsilon_{p}$ and $\varepsilon_{c}$ were reduced with the decreasing strain rate and rising temperature. The $\varepsilon_{p}$ is associated with deformation temperature (*T*) and strain rate ($\overset{\cdot }{\varepsilon }$), and the relationship among $\varepsilon_{p}$, *T* and $\dot{\varepsilon }$ can be given as [10,21]:(1)$$\varepsilon_{p}=ad_{0}^{n_{1}}\dot{\varepsilon }{}^{m_{1}}\exp(\frac{Q_{1}}{RT})$$
where $d_{0}^{}$ is the initial grain size of the TB8 titanium alloy after heat treatment. a, $n_{1}$ and $m_{1}$ are material constants and the value of R is 8.314 J·K^−1^·mol^−1^. $Q_{1}$ represents the deformation activation energy. In this paper, the value of $d_{0}^{}$ is the same for all deformed samples. The effect of the $d_{0}^{}$ value on the peak or critical strain can be ignored. Thus, Equation (1) can be simplified to Equation (2):(2)$$\varepsilon_{p}=a\dot{\varepsilon }{}^{m_{1}}\exp(\frac{Q_{1}}{RT})$$

In order to compute the values of a, $m_{1}$ and $Q_{1}$, it could be beneficial to take the natural logarithm on two sides of Equation (2). Equation (2) can be expressed by means of Equation (3):(3)$$\ln\varepsilon_{p}=\ln a+m_{1}\ln\dot{\varepsilon }+\frac{Q_{1}}{RT}$$

Based on the linear relationship between $\ln\varepsilon_{p}$ and $\ln\dot{\varepsilon }$, the value of $m_{1}$ can be obtained to be 0.212 by linearly fitting the data in Figure 11. Similarly, the $Q_{1}$ was also calculated from Figure 12 to be 29130 J/mol. Ultimately, the value of a was computed as about 0.00214 by means of substituting $m_{1}$ and $Q_{1}$ into Equation (3). Thus, Equation (2) can be given as:(4)$$\varepsilon_{p}=2.14\times 10^{-3}\dot{\varepsilon }{}^{0.212}\exp(\frac{29130}{RT})$$

The relationship between $\varepsilon_{c}$ and $\varepsilon_{p}$ can be described by [21,22]
(5)$$\varepsilon_{c}=\beta \varepsilon_{p}$$
where β is a material constant. The value of β can be obtained from Figure 13 as 0.863 by linear fitting. By combining Equations (4) with Equation (5), $\varepsilon_{c}$ can be characterized as:(6)$$\varepsilon_{c}=1.85\times 10^{-3}\dot{\varepsilon }{}^{0.212}\exp(\frac{29130}{RT})$$

### 3.3. Kinetics Model of DRX

During the hot working of metal materials, the DRX behavior, including nucleation and growth, is dramatically dependent on the density of dislocation. In general, the nucleation of DRX preferentially occurs near to the grain boundaries of deformed grains, with a high strain gradient [23], and then grows with the driving force induced by the difference of dislocation density on either side of the grain boundaries of deformed grains [24]. The correlation between the $X_{DRX}$ and the strain (ε) can written by means of the JMAK equation [4,6,10,25,26]:(7)$$X_{DRX}=1-\exp\left[-k\times {\left(\frac{\varepsilon -\varepsilon_{c}}{\varepsilon_{0.5}}\right)}^{n}\right]$$
where k and n are the Avrami material constants. The $\varepsilon_{0.5}$ is the strain corresponding to the occurrence of 50% $X_{DRX}$. During hot working, incomplete DRX is obtained when the flow stress (σ) does not reach the stable flow stress. The relationship between σ and $X_{DRX}$ can be given as [25,26]:(8)$$X_{DRX}=\frac{\sigma_{p}-\sigma }{\sigma_{p}-\sigma_{ss}}$$
where $\sigma_{p}$ and $\sigma_{ss}$ are defined as the peak stress and the steady flow stress, respectively. Therefore, when the $X_{DRX}$ reaches 50%, the corresponding stress ($\sigma_{0.5}$) can be rewritten as:(9)$$\sigma_{0.5}=\frac{1}{2}(\sigma_{p}+\sigma_{ss})$$

According to the flow stress data of TB8 titanium alloy, as well as the value of $X_{DRX}$ calculated by the optical photographs after hot deformation, the $\sigma_{0.5}$ can be achieved by Equation (9). Afterwards, the $\varepsilon_{0.5}$ corresponding to $\sigma_{0.5}$ is also determined under different deformation conditions. The relationship among $\varepsilon_{0.5}$, $\dot{\varepsilon }$ and *T* can be described as [27]:(10)$$\varepsilon_{0.5}=b\dot{\varepsilon }{}^{m_{2}}\exp(\frac{Q_{2}}{RT})$$
where b and $m_{2}$ are material constants. $Q_{2}$ is the recrystallization activation energy. The values of b, $m_{2}$ and $Q_{2}$ can calculated to be 0.044, 0.288 and 30016 J/mol by a method similar to Equation (3), respectively. Thus, Equation (10) can be written as follows:(11)$$\varepsilon_{0.5}=4.40\times 10^{-2}\dot{\varepsilon }{}^{0.288}\exp(\frac{30016}{RT})$$

To compute the values of k and n in Equation (7), we take the natural logarithm on two sides of Equation (7), and then Equation (7) can be written by means of Equation (12):(12)$$\ln[-\ln(1-X_{DRX})]=\ln k+n\ln[(\varepsilon -\varepsilon_{c})/\varepsilon_{0.5}]$$

The average values of k and n can be obtained, from Figure 14, to be 0.172 and 1.514 by linear fitting, respectively. As such, the kinetic model of DRX can be exhibited as:(13)$$X_{DRX}=1-\exp\left[-0.172\times {\left(\frac{\varepsilon -\varepsilon_{c}}{\varepsilon_{0.5}}\right)}^{1.514}\right]$$

Figure 15 shows the variation in $X_{DRX}$ with true strain. It is revealed that $X_{DRX}$ increases dramatically with the increase in true strain in the form of a classical S-curve under given hot processing conditions. The rising temperature and reducing strain rate noticeably promote the occurrence of DRX during hot working. To further analyze the effect of the deformation parameters on $d_{DRX}$, $d_{DRX}$ as a function of the Z parameter ($Z=\dot{\varepsilon }\exp(\frac{Q}{RT})$) was characterized by [28]:(14)$$d_{DRX}=CZ^{m_{3}}$$
where C and $m_{3}$ are material constants. Equation (14) can be rewritten into the form of Equation (15) by taking the natural logarithm on both sides of Equation (14):(15)$$\ln d_{DRX}=\ln C+m_{3}\ln Z$$

The values of C and $m_{3}$ can be respectively determined to be 5233.68 and −0.279 by the linear fitting of the data in Figure 16. As such, Equation (14) can be expressed as follows:(16)$$d_{DRX}=5.23\times 10^{3}Z^{-0.279}$$

### 3.4. FE Simulation of DRX Behavior

According to the flow curve gained from experimental tests and the above developed models for the DRX of TB8 titanium alloy, the isothermal forging process of the cylindrical samples was simulated under various hot working parameters with the DEFORM 3D software. In the process of establishing the FE model, the elastic deformation of the workpiece can usually be ignored under the high-temperature plastic deformation condition. Therefore, the workpiece was considered as a plastic body, while the tools were regarded as rigid bodies [25,29,30]. So as to reducing the time of FE simulation and improve the simulation accuracy, the model of FE simulation was constructed using half of the cylinder for the cylindrical samples with symmetry. The sample for FE simulation was segmented into tetrahedral elements. The mesh number and the node quantity were set as 17,398 and 3983, respectively. The top die was set so as to move along the central axis of the deformed samples. The friction type between the sample and the die is usually considered as shear friction [31]. The value of the friction coefficient was set to be 0.3. In order to keep consistency between the experimental results obtained from Gleeble compression tests and the simulation results gained from the FE simulation, the temperatures of the samples, die and compressed environment were all considered as the same as the experimental temperature during the process of establishing the FE model. 

Table 3 lists the distribution of effective strain for TB8 titanium alloys deformed at different processing parameters. It can be seen that the strain distribution is non-uniform during hot working. The minimum value of the effective strain was located in the center of the top and bottom of the cylinder sample, while the maximum value of the effective strain was distributed in the center of the section of samples along the forging axis under a given deformation condition. A larger effective strain easily results in the generation of a high density of dislocation, which is favorable for the nucleation of dynamic recrystallization. Thus, the dynamic recrystallization was more easily induced in the center of the section of samples along the forging axis than it was in other regions. Moreover, it can also be seen from Table 3 that the distribution of effective strain was significantly affected by the processing parameters. The uniformity of strain distribution in the center of the deformed samples was increased with augmenting the temperature and reducing the strain rate.

Table 4 displays the distribution in the $X_{DRX}$ modeled by FEM at a strain of 0.8. The distribution in the $X_{DRX}$ presents the same variation tendency as the distribution of effective strain. The maximum value in the volume fraction of DRX was also located in the center of the section of samples along the forging axis under a given deformation condition. The phenomenon of inhomogeneous distribution in the $X_{DRX}$ is attributed to the shear friction and thermal exchange at the top and bottom of the cylinder sample during hot deformation. Moreover, it is also seen from Table 4 that the rising temperature and reducing strain rate facilitated an increase in the $X_{DRX}$ at a given strain of 0.8. 

Figure 17 and Figure 18 present a comparison of the $d_{DRX}$ and $X_{DRX}$ in the central regions of the samples between the experimental and FEM results. It can be seen from the results that all display the same variation tendency for $d_{DRX}$ and $X_{DRX}$, which increased with the increase in temperature and the reduction in strain rate. Generally, a high strain rate leads more easily to the generation of dislocation and the increase in dislocation density, which is beneficial to the formation of DRX during hot working. Nevertheless, in this present work, it can be noted that the grain size and volume fraction of DRX at a lower strain rate are obviously higher than those at higher strain rates. This indicates that enough time, owing to a lower strain rate, is more helpful in the growth of DRX grains for TB8 titanium alloys.

So as to confirm the accuracy of the FEM results obtained from the derived microstructure evolution models, the correlation coefficient ($R^{2}$) and the average absolute relative error (∆) value between the experimental and FEM results were calculated by the following formula [25]:(17)$$R^{2}=\frac{\sum_{i=1}^{N}\left(X_{E}-{\bar{X}}_{E})(X_{s}-{\bar{X}}_{s}\right)}{\sqrt{\sum_{i=1}^{N}{\left(X_{E}-{\bar{X}}_{E}\right)}^{2}{\left(X_{s}-{\bar{X}}_{s}\right)}^{2}}}$$
(18)$$\Delta (\%)=\frac{1}{N}\sum_{i=1}^{N}\left|\frac{X_{E}-X_{s}}{X_{E}}\right|\times 100$$
where $X_{E}$ is the experimental volume fraction of DRX and $X_{s}$ is the simulation volume fraction of DRX derived from the developed microstructure evolution models. ${\bar{X}}_{E}$ and ${\bar{X}}_{s}$ are the average values of $X_{E}$ and $X_{s}$, respectively. In addition, *N* is the total amount of data employed in this study. It can be observed from Figure 19 and Figure 20 that a good correlation between the experimental and simulation results was obtained. The value of R was 0.973 and 0.994 for the $d_{DRX}$ and $X_{DRX}$, respectively. This suggests that the DRX behavior of TB8 titanium alloy can be better simulated by using DEFORM-3D software. The ∆ value was calculated through a term by term comparison of the relative error. In this paper, the volume fractions of DRX at 850 °C/1 s^−1^, 900 °C/1 s^−1^, 950 °C/1 s^−1^ and 850 °C/0.1 s^−1^ were lower than 5%, which can be ignored in the calculation processing of the ∆ value. Thus, the ∆ value for the $d_{DRX}$ and $X_{DRX}$ is calculated to be 3.2% and 17.0%, respectively, indicating that the established FEM model for TB8 titanium alloy during hot working presents a good prediction capability.

## 4. Conclusions

In the present paper, the DRX behavior of TB8 titanium alloy during hot working was investigated by the experiment and FEM method. The main conclusions are as follows:(1)The DRX behavior of TB8 titanium alloys was drastically influenced by the processing parameters. The rising deformation temperature and reducing strain rate resulted in an increase in the $d_{DRX}$ and $X_{DRX}$;(2)Based on the true stress–strain curves obtained from experimental tests, the $X_{DRX}$ and $d_{DRX}$ kinetics models of DRX were established, which were expressed as $\left\{\begin{array}{c} X_{DRX}=1-\exp\left[-0.172\times {\left(\frac{\varepsilon -\varepsilon_{c}}{\varepsilon_{0.5}}\right)}^{1.514}\right]\\ d_{DRX}=5.23\times 10^{3}Z^{-0.279} \end{array}\right\}$(3)In terms of the true stress–strain curve gained from isothermal tests and the developed models for the DRX of TB8 titanium alloy, the isothermal forging process of the cylindrical samples was simulated by the DEFORM-3D software. The distributions of effective strain and DRX volume fraction were analyzed for TB8 titanium alloys during hot deformation. The maximum values of the effective strain and $X_{DRX}$ were distributed in the center of the section of samples along the forging axis under a given deformation condition;(4)The value of the correlation coefficient was respectively 0.973 and 0.994 for the $d_{DRX}$ and $X_{DRX}$ between the experimental and FE simulation results, while the average absolute relative error value for the $d_{DRX}$ and $X_{DRX}$ was respectively 3.2% and 17.0%, which presents the good prediction capabilities of the established FEM model for TB8 titanium alloy during hot working.

## Figures and Tables

**Figure 1 materials-13-04429-f001:**
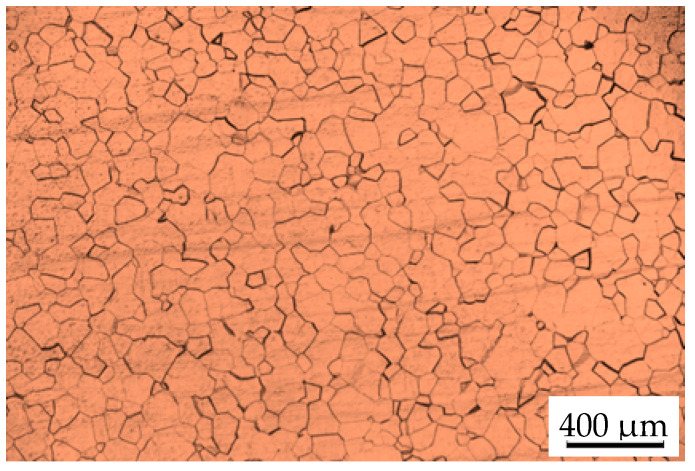
Microstructure of the samples after solution-treatment at 900 °C for 30 min.

**Figure 2 materials-13-04429-f002:**
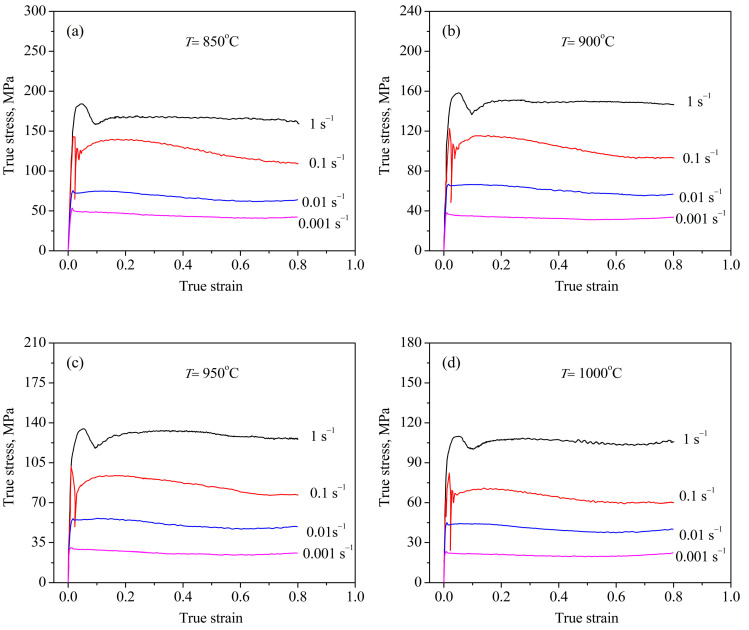
(**a**–**d**) Stress–strain curves of TB8 titanium alloys at different deformation conditions.

**Figure 3 materials-13-04429-f003:**
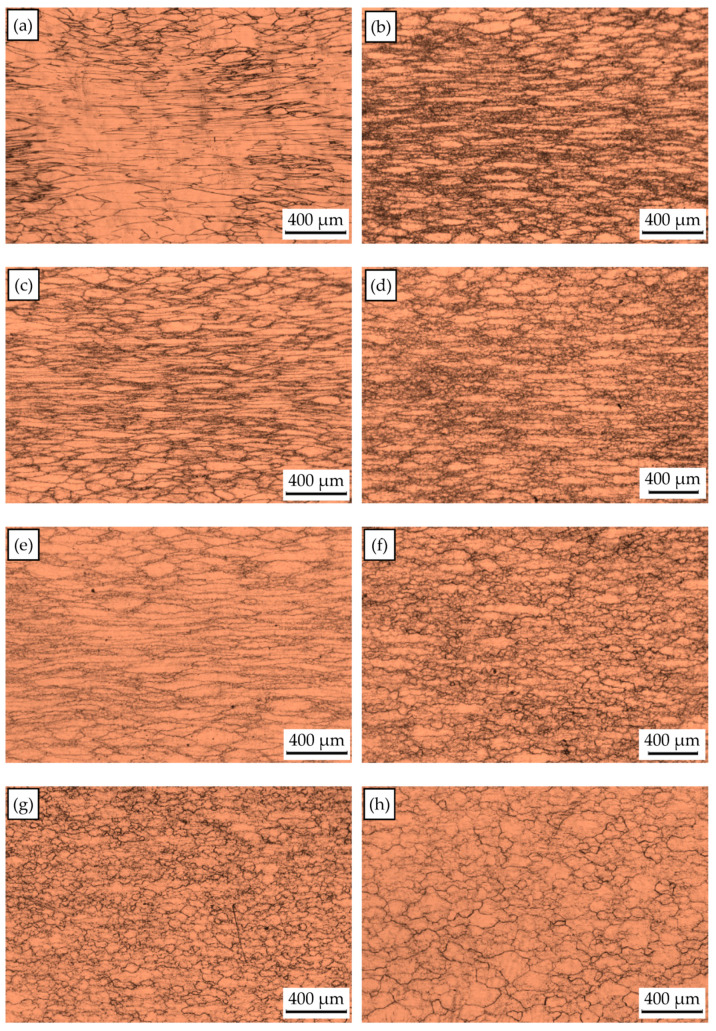
Optical microstructures of the deformed samples at a strain of 0.8 and deformation conditions of (**a**) 850 °C/1 s^−1^, (**b**) 850 °C/0.01 s^−1^, (**c**) 900 °C/0.1 s^−1^, (**d**) 950 °C/0.01 s^−1^, (**e**) 1000 °C/0.1 s^−1^, (**f**) 1000 °C/0.01 s^−1^, (**g**) 900 °C/0.001 s^−1^ and (**h**) 1000 °C/0.001 s^−1^.

**Figure 4 materials-13-04429-f004:**
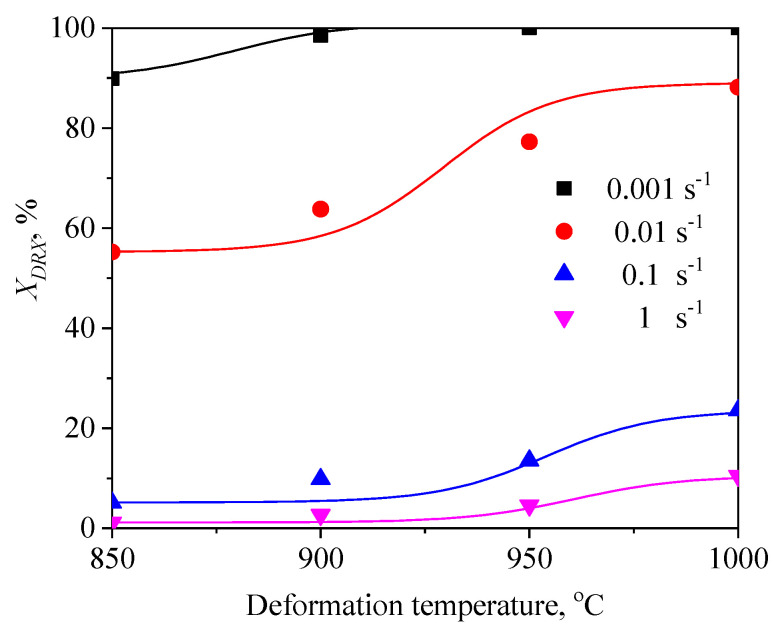
Variation in the volume fraction of recrystallized grains ($X_{DRX}$) with deformation parameters.

**Figure 5 materials-13-04429-f005:**
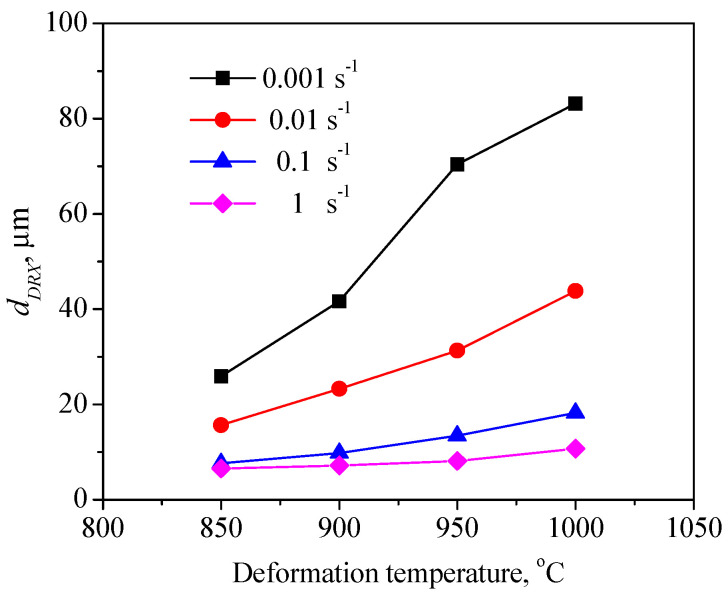
The grain size of recrystallized grains ($d_{DRX}$) as a function of deformation parameters.

**Figure 6 materials-13-04429-f006:**
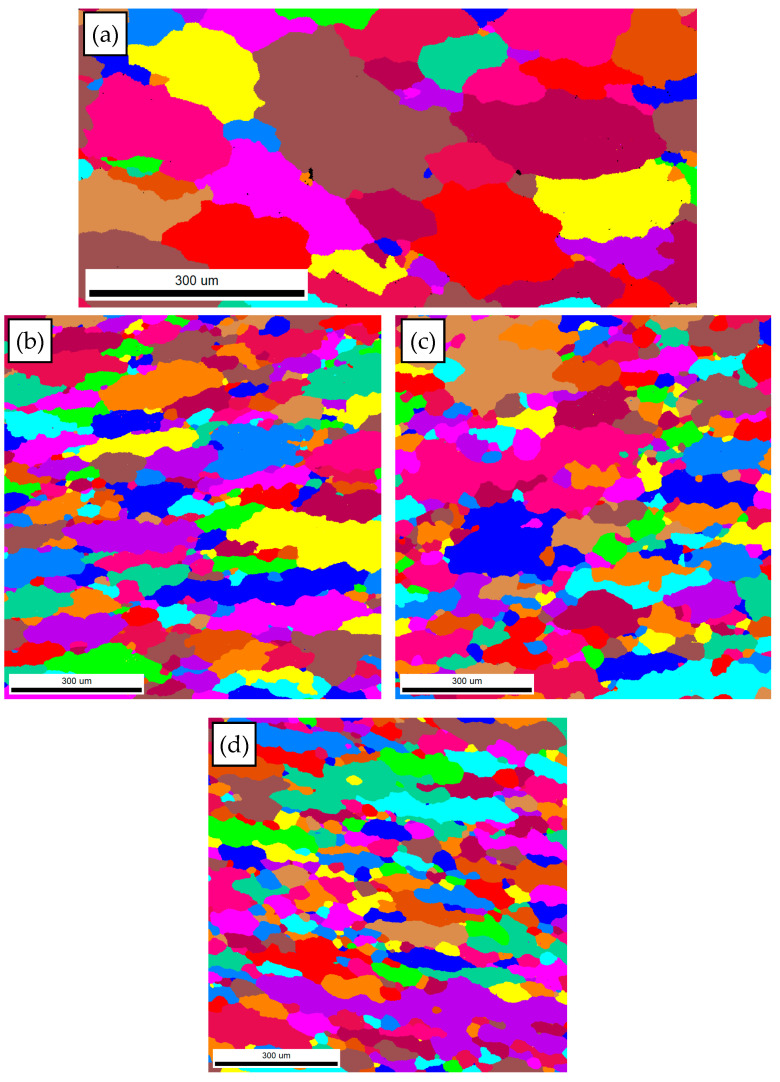
EBSD grain maps of the TB8 titanium alloy deformed at different strains of (**a**) 0.2, (**b**) 0.4, (**c**) 0.6 and (**d**) 0.8.

**Figure 7 materials-13-04429-f007:**
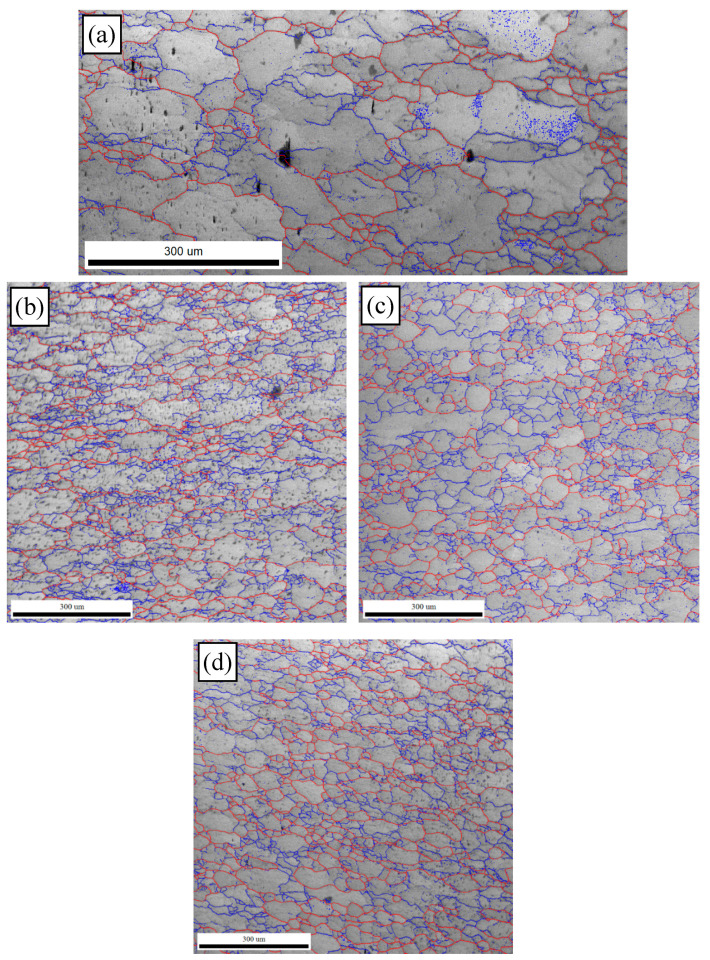
EBSD IQ maps overlaid with grain boundaries of the TB8 titanium alloy deformed at different strains of (**a**) 0.2, (**b**) 0.4, (**c**) 0.6 and (**d**) 0.8.

**Figure 8 materials-13-04429-f008:**
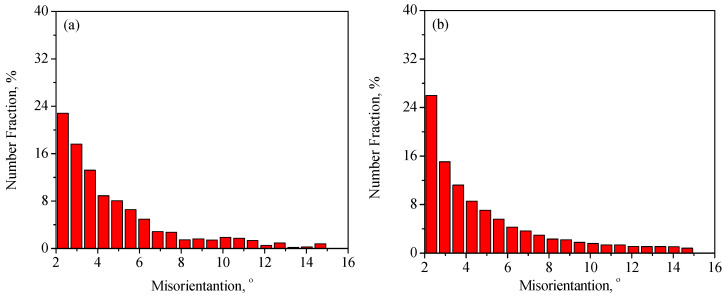
Distribution histograms of misorientation angles in the range 2–15° for the solution-treated samples deformed at different strains of (**a**) 0.2, (**b**) 0.4, (**c**) 0.6 and (**d**) 0.8.

**Figure 9 materials-13-04429-f009:**
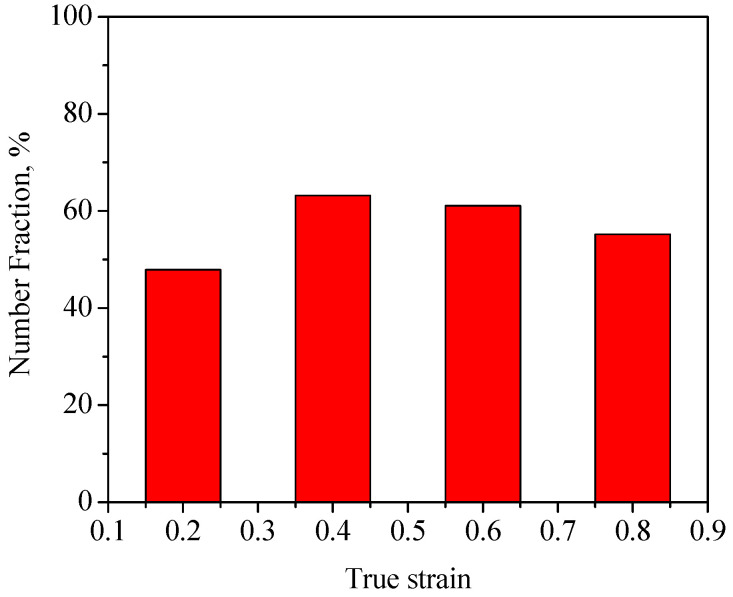
Variation of the number fraction of the misorientation angles in the range 2–15° with true strain.

**Figure 10 materials-13-04429-f010:**
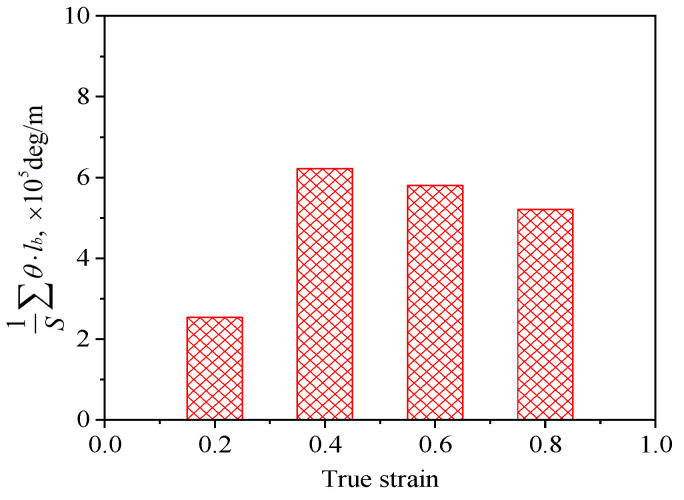
Variation of the microstructural parameter ($\frac{1}{S}\sum \theta \times lb$) with true strain for the solution-treated samples deformed at 900 °C and 10^−3^ s^−1^.

**Figure 11 materials-13-04429-f011:**
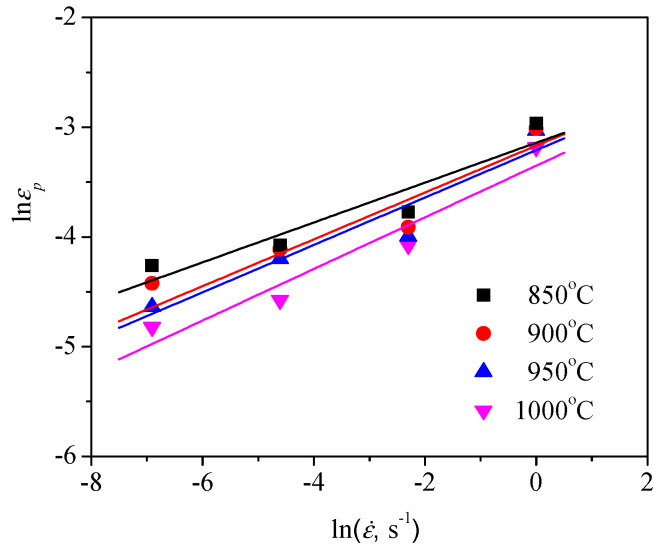
Relationship between the peak strain ($\varepsilon_{p}$) and strain rate ($\dot{\varepsilon }$).

**Figure 12 materials-13-04429-f012:**
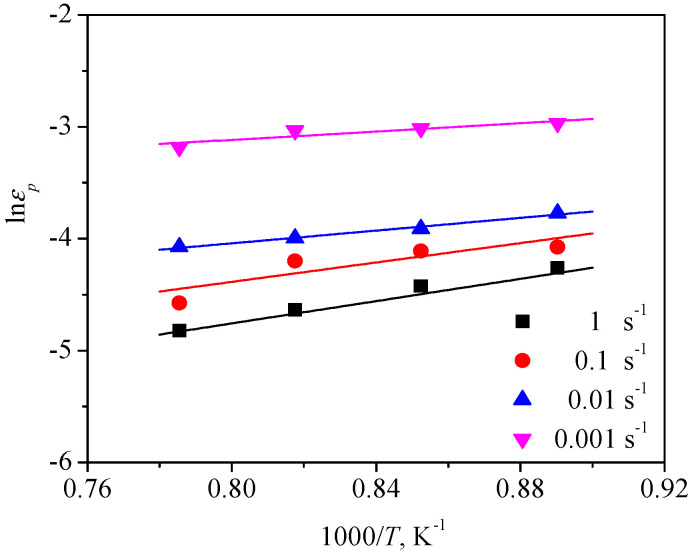
Relationship between the peak strain ($\varepsilon_{p}$) and the temperature (*T*).

**Figure 13 materials-13-04429-f013:**
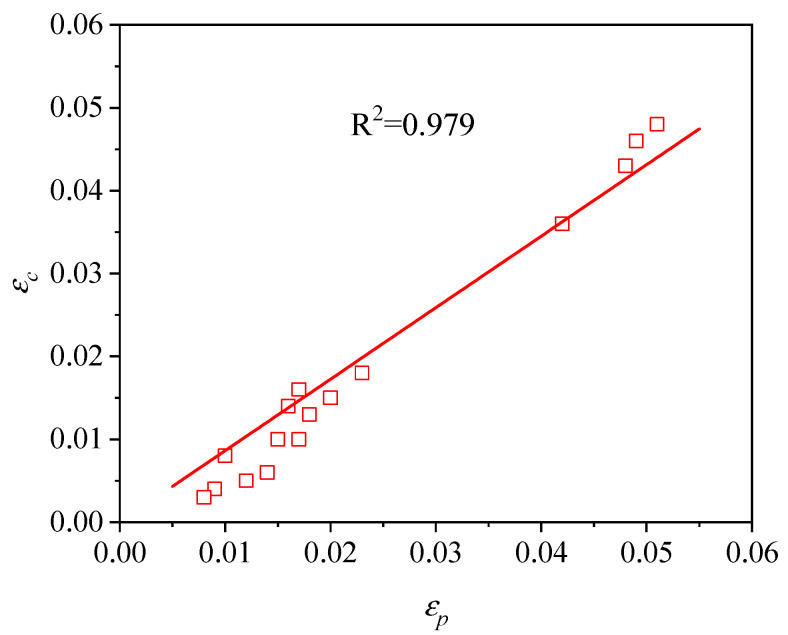
Relationship between the critical strain ($\varepsilon_{c}$) and peak strain ($\varepsilon_{p}$).

**Figure 14 materials-13-04429-f014:**
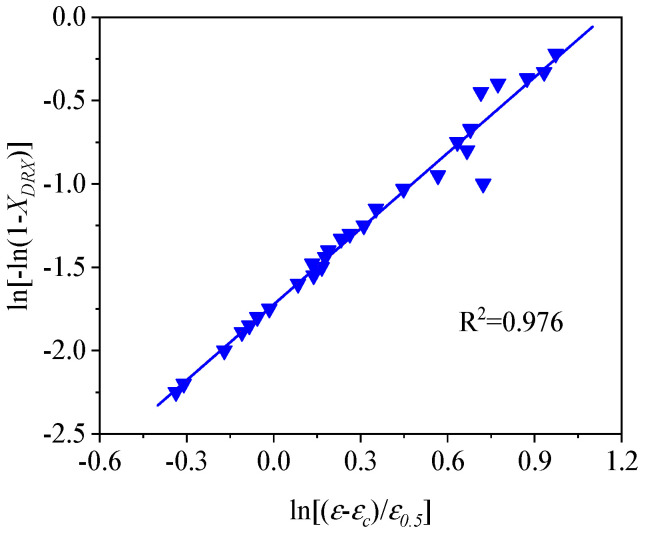
Relationship between $\ln[-\ln(1-X_{DRX})]$ and $\ln[(\varepsilon -\varepsilon_{c})/\varepsilon_{0.5}]$.

**Figure 15 materials-13-04429-f015:**
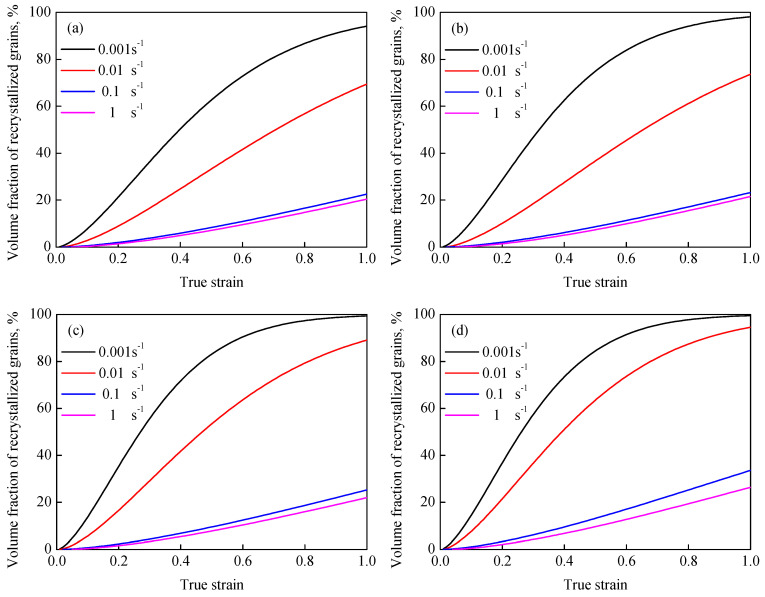
Variation in the $X_{DRX}$ with true strain at different temperatures of (**a**) 850 °C, (**b**) 900 °C, (**c**) 950 °C and (**d**) 1000 °C.

**Figure 16 materials-13-04429-f016:**
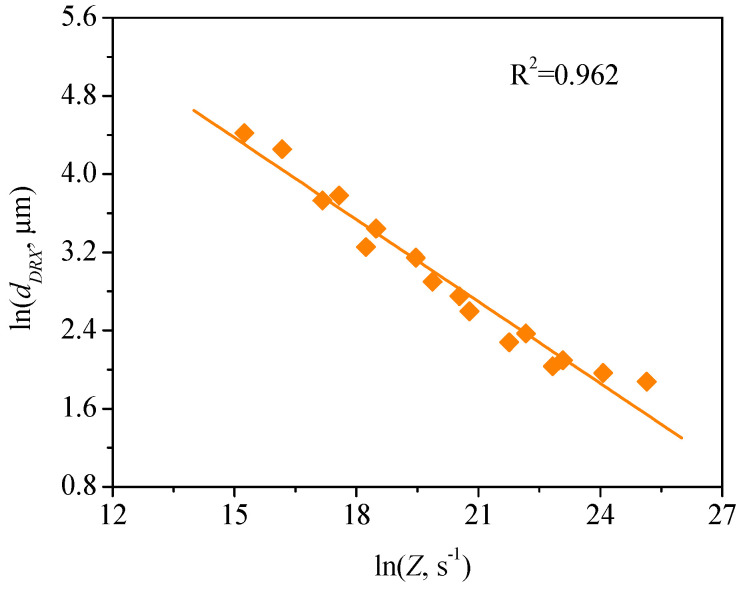
Relationship between $d_{DRX}$ and Zener –Hollomon parameter (*Z*).

**Figure 17 materials-13-04429-f017:**
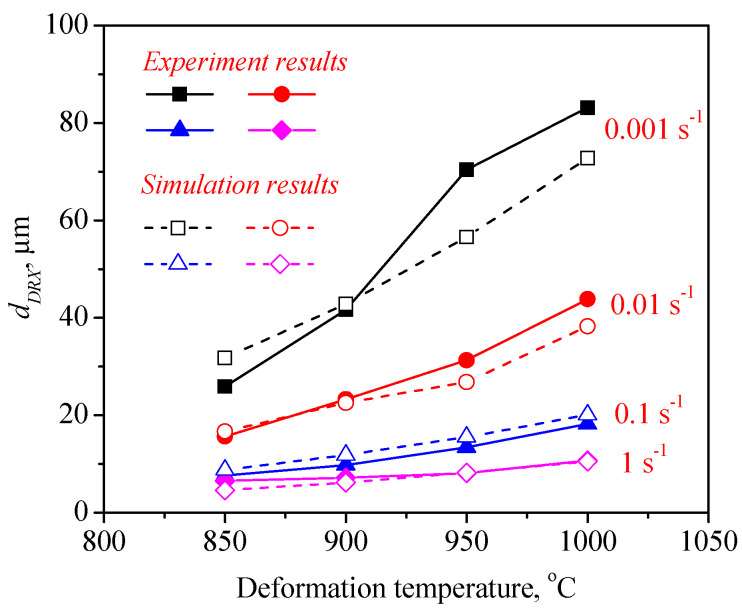
A comparison between the experimental and simulation results for the $d_{DRX}$.

**Figure 18 materials-13-04429-f018:**
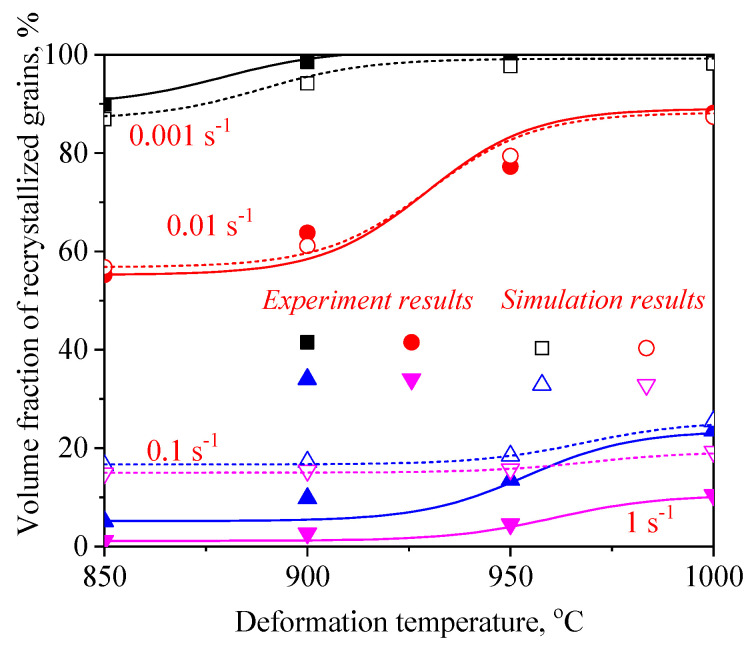
A comparison between the experimental and FE simulation results for the $X_{DRX}$ after hot working.

**Figure 19 materials-13-04429-f019:**
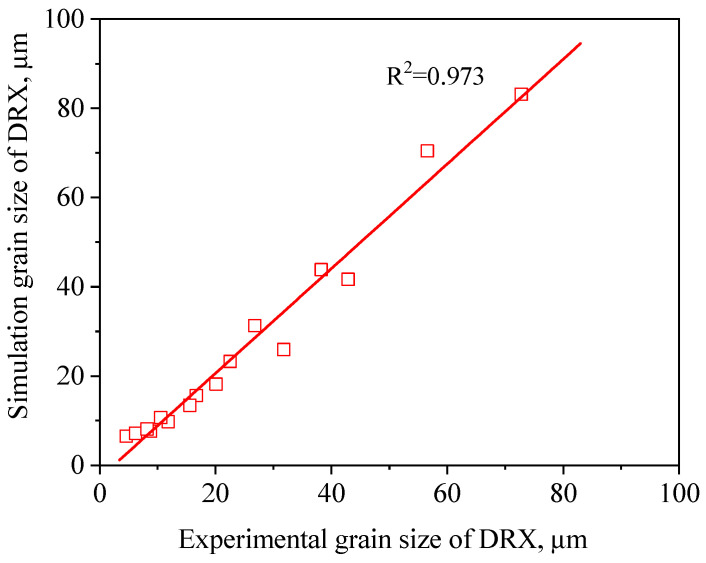
Correlation between the simulation and experimental $d_{DRX}$ for TB8 titanium alloys after hot deformation.

**Figure 20 materials-13-04429-f020:**
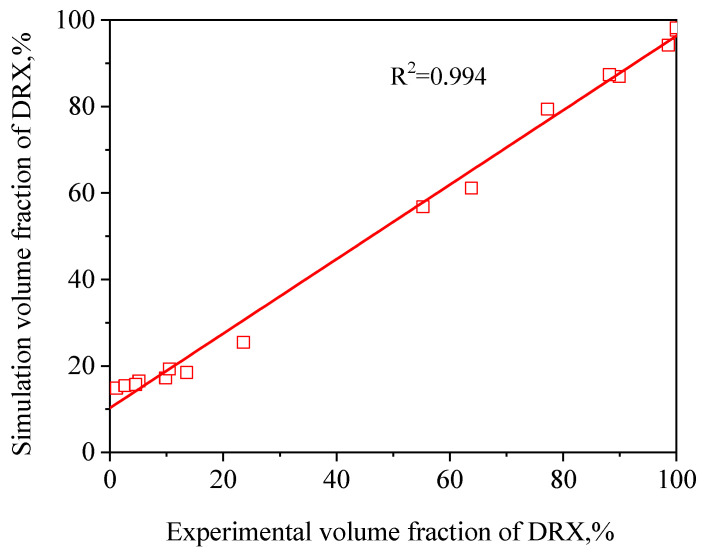
Correlation between the simulation and experimental $X_{DRX}$ for TB8 titanium alloys after hot deformation.

**Table 1 materials-13-04429-t001:** Chemical composition of the TB8 titanium alloys (wt. %).

Mo	Al	Nb	Si	Fe	C	N	O	H	Ti
14.5	2.9	2.85	0.19	0.07	0.02	0.02	0.09	0.002	Bal

**Table 2 materials-13-04429-t002:** The values of $\varepsilon_{p}$ and $\varepsilon_{c}$ at different deformation parameters.

Temperature	$\varepsilon_{p}$				$\varepsilon_{c}$			
	1 s^−1^	0.1 s^−1^	0.01 s^−1^	0.001 s^−1^	1 s^−1^	0.1 s^−1^	0.01 s^−1^	0.001 s^−1^
850 °C	0.051	0.023	0.017	0.014	0.048	0.018	0.016	0.006
900 °C	0.049	0.02	0.016	0.012	0.046	0.015	0.014	0.005
950 °C	0.048	0.018	0.015	0.009	0.043	0.013	0.010	0.004
1000 °C	0.042	0.017	0.010	0.008	0.036	0.01	0.008	0.003

**Table 3 materials-13-04429-t003:** The distribution of effective strain modeled by FEM for TB8 titanium alloy deformed at the strain of 0.8.

Strain Rate/s^−1^	Temperature/°C				Effective Strain
	850	900	950	1000	
0.001	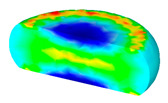	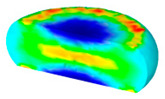	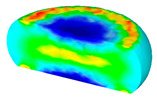	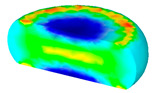	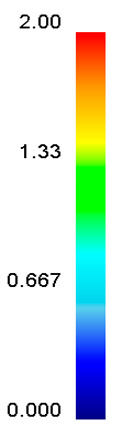
0.01	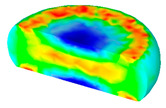	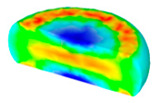	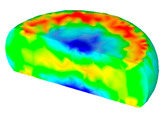	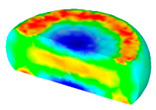	
0.1	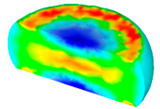	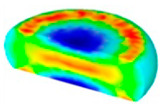	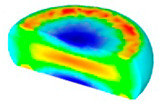	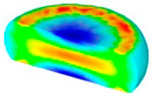	
1	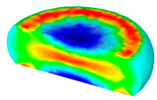	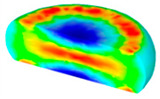	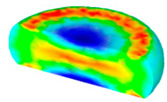	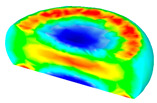	

**Table 4 materials-13-04429-t004:** The distribution in the $X_{DRX}$ by FEM for TB8 titanium alloy deformed at a strain of 0.8.

Strain Rate/s^−1^	Temperature/°C				X_DRX_
	850	900	950	1000	
0.001	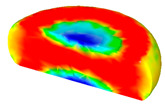	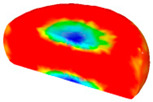	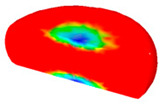	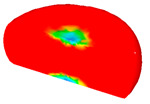	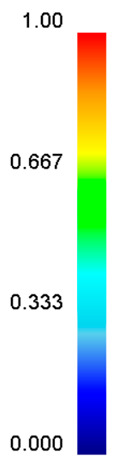
0.01	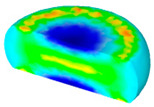	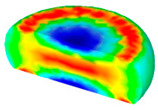	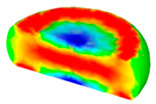	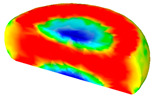	
0.1	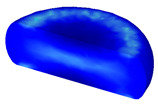	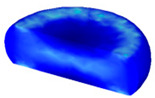	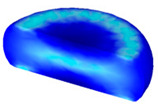	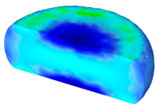	
1	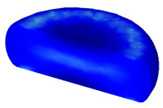	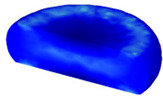	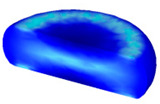	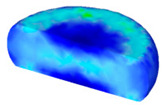	

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
