# Peer review of "Simulation and Experimental Study of Dynamical Recrystallization Kinetics of TB8 Titanium Alloys"

_materials, 2020, doi:10.3390/ma13194429_

Round 1
Reviewer 1 Report
The reviewed article ”Simulation and experimental study of dynamical recrystallization kinetics of TB8 titanium alloys” is interesting and enrich the knowledge regarding the dynamic recrystallization behavior of TB8 – a new type of metastable β titanium alloy. The results of carried out experiments are presented in a good scientific level. The comprehensive description of obtained results is given in a clear and detailed. The great advantage of the described research works is the support of the experiments with numerical simulations. I highly rate the presented work nevertheless, I ask the authors to refer to the following minor remarks:
- Page 1, Line: 39: Please delete the unnecessary free space between ”technology”, and “the”.
- Page 2, Line: 83: Should be ”815 °C” rather than ”815°C”. Please carefully check and correct this mistake in the entire manuscript.
- Page 3, Line: 88: Please provide the type and manufacturer of the instrument with which the image of the TB8 titanium microstructure was obtained (Leica DMI5000M optical microscope?).
- Page 3, Line: 90: Should be ”ø8×12 mm” rather than ”Φ8×12 mm”.
- Page 3, Line: 94: Should be ”20 °C/s” rather than ”20ºC/s”. Please carefully check and correct this mistake in the entire manuscript.
- Page 3, Line: 111: Should be ”Figure 2” rather than ” 2”. Term ”Figure” is used in the main text of the manuscript, whereas ”Fig.” only in brackets. Please carefully check and correct this mistake in the entire manuscript.
- Page 5, Line: 148: Please remove the unnecessary parenthesis of ((a) 850 1 s-1.
- Page 6, Line: 164: Should be ”(2° ~ 15°)” rather than ”(2o ~ 15o)”. Please carefully check and correct this mistake in the entire manuscript.
- Page 7, Line: 181: Please check font size.
- Page 7/8: Figure 6 with its caption must be at one page.
- Page 10, Line: 195: You wrote ”Histograms” as a capital letter.Why?
- Page 10, Line: 203: The description of Figure 9 is given in subsection 3.1, but the figure is in subsection 3.2. Why?
- The authors use many symbols and acronyms. I suggest consider inserting their explanation in the Nomenclature at the end of the work (after the Conflicts of Interest).
- Please carefully check English language and grammar.

Reviewer 2 Report
Dear Authors,
Comments and suggestions are attached in the file "Comments".
Reviewer
